# Asymmetric Factorized Bilinear Operation for Vision Transformer

**Junjie Wu**[1] **Qilong Wang**[1,*] **Jiangtao Xie**[2] **Pengfei Zhu**[1] **Qinghua Hu**[1]
[1]Tianjin University   [2]Dalian University of Technology
{wjj_, qlwang, zhupengfei, huqinghua}@tju.edu.cn, jiangtaoxie@mail.dlut.edu.cn

## Abstract

As a core component of Transformer-like deep architectures, a feed-forward network (FFN) for channel mixing is responsible for learning features of each token. Recent works show channel mixing can be enhanced by increasing computational burden or can be slimmed at the sacrifice of performance. Although some efforts have been made, existing works are still struggling to solve the paradox of performance and complexity trade-offs. In this paper, we propose an Asymmetric Factorized Bilinear Operation (AFBO) to replace FFN of vision transformer (ViT), which attempts to efficiently explore rich statistics of token features for achieving better performance and complexity trade-off. Specifically, our AFBO computes second-order statistics via a spatial-channel factorized bilinear operation for feature learning, which replaces a simple linear projection in FFN and enhances the feature learning ability of ViT by modeling second-order correlation among token features. Furthermore, our AFBO presents two structured-sparsity channel mapping strategies, namely Grouped Cross Channel Mapping (GCCM) and Overlapped Cycle Channel Mapping (OCCM). They decompose bilinear operation into grouped channel features by considering information interaction between groups, significantly reducing computational complexity while guaranteeing model performance. Finally, our AFBO is built with GCCM and OCCM in an asymmetric way, aiming to achieve a better trade-off. Note that our AFBO is model-agnostic, which can be flexibly integrated with existing ViTs. Experiments are conducted with twenty ViTs on various tasks, and the results show our AFBO is superior to its counterparts while improving existing ViTs in terms of generalization and robustness.

## 1 Introduction

In recent years, transformer-like architectures have attracted a mass of research interests and achieved remarkable performance in various computer vision tasks (Dosovitskiy et al., 2021; Fang et al., 2021; Strudel et al., 2021; Neimark et al., 2021). As one of core components, a feed-forward network (FFN) with two fully-connected layers is generally used as channel mixer to learn token features, which is proven to significantly influence performance of vision transformer (ViT) (Dong et al., 2021; Yu et al., 2022). Intuitively, a two-layer FFN is a concise yet naive learning scheme that fails to fully consider the rich information lying in token features, which generally achieves sub-optimal solutions in terms of both efficiency and effectiveness (Fang et al., 2024; Xu et al., 2024; Sridhar et al., 2023). However, existing works pay less attention to improving FFN module of ViT compared to the modifications on self-attention mechanism (i.e., another core component of ViT) (Huang et al., 2021; Liu et al., 2021; Kitaev et al., 2020; Graham et al., 2021; Srinivas et al., 2021; Touvron et al., 2021b).

Recently, some works have made attempts to improve channel mixing module of transformer-like architectures (Fang et al., 2024; Xu et al., 2024; Sridhar et al., 2023; Li et al., 2021). As a pioneer work, Shazeer (Shazeer, 2020) introduces to exploit Gated Linear Units (GLU) (Dauphin et al., 2017) to improve FFN of transformer in natural language processing tasks. Subsequently, EVA-02 (Fang et al., 2024) extends GLU to ViT by employing a SiLU activation (Hendrycks & Gimpel, 2016).

---

*Corresponding author.

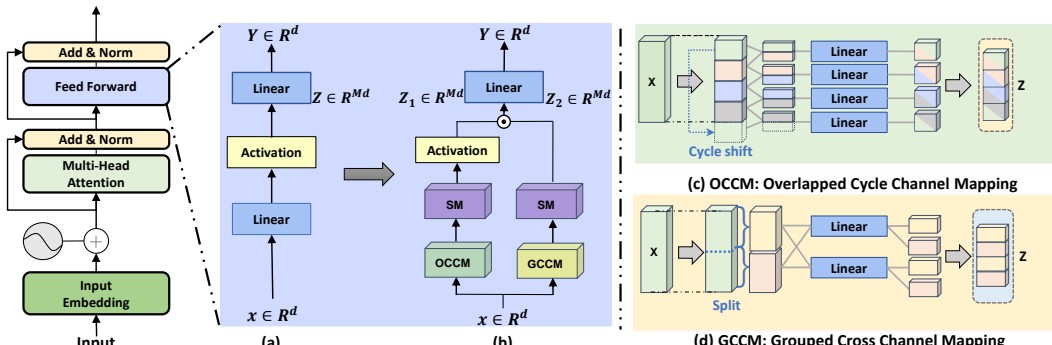

Figure 1: Comparison of (a) original FFN with (b) our proposed Asymmetric Factorized Bilinear Operation (AFBO) for ViT. Specifically, our AFBO computes second-order statistics via a spatial-channel factorized bilinear operation (SCFBO) for feature abstraction instead of the first linear projection in FFN. Particularly, our SCFBO first approximates second-order statistics via a rank-1 decomposition, and further decomposes mapping operations into Spatial Modeling (SM) and Channel Mapping (CM). For better performance and complexity trade-off, our AFBO is constructed by two structured-sparsity channel mapping strategies (i.e., (c) Grouped Cross Channel Mapping and (d) Overlapped Cycle Channel Mapping) in an asymmetric manner. The details refer to Sec. 3.2.

To guarantee computational efficiency, existing GLU variants (Shazeer, 2020; Fang et al., 2024) generally require to reduce the hidden dimension of FFN, which potentially limits performance of ViT (Noshad et al., 2019; Sridhar et al., 2023). Besides, some researchers propose to use some extra modules inheriting from convolutional neural networks (CNNs) to enhance FFN of ViT (Li et al., 2021; Guo et al., 2022a; Zhou et al., 2022; Cao et al., 2023), such as depth-wise (DW) convolution, channel attention (Hu et al., 2020; Wang et al., 2020; Zhou et al., 2022), inverted residual FFN (Sandler et al., 2018) and GRN (Liu et al., 2022; Woo et al., 2023). Although these extra modules can improve performance of ViT, they undoubtedly increase computational cost, especially for large models. In contrast, some recent works (Xu et al., 2024; Sridhar et al., 2023) focus on slimming FFN to reduce computational cost of transformer-like architectures by carefully decreasing hidden dimension to reduce computational cost or employing a block diagonal linear projection regularized by a channel covariance attention. Despite the aforementioned slimming methods can reduce computational cost of FFN, they are limited to improve performance of ViT. Therefore, existing FFN variants of ViT still struggle to solve the paradox of performance and complexity trade-off.

To address the above issue, we propose a novel Asymmetric Factorized Bilinear Operation (AFBO) to efficiently explore and exploit rich information of token features, which replaces FFN of Transformer for achieving better performance and complexity trade-off. Specifically, motivated by the success of second-order modeling in improving deep CNNs (Lin et al., 2015; Gao et al., 2016; Li et al., 2017; Wang et al., 2017; 2021a), our AFBO presents a Spatial-Channel Factorized Bilinear Operation (SCFBO) to efficiently compute second-order statistics of token features, which replaces the first linear projection in the FFN for feature learning. Particularly, our SCFBO first approximates second-order statistics via a rank-1 decomposition (Gao et al., 2016; Li et al., 2017; Mnih & Hinton, 2007), which is further decomposed into spatial modeling and channel mapping. For spatial modeling, we capture spatial correlation in local neighborhoods using various methods, including local pooling, DW convolution, and spatial attention. For channel mapping, two point-wise convolutions followed by a dot product are used to model the channel correlation of each token feature. As such, our AFBO enhances the feature learning ability of Transformer by modeling second-order correlation among token features. Since channel mapping usually leads to high computational complexity especially with large hidden dimension, our AFBO presents two structured-sparsity channel mapping strategies, namely Grouped Cross Channel Mapping (GCCM) and Overlapped Cycle Channel Mapping (OCCM), which divide channel features into several groups and perform bilinear operation by considering information interaction between groups. Based on the proposed structured-sparsity channel mappings, we can construct several AFBO variants that significantly reduce parameters and FLOPs while maintaining high performance. Particularly, OCCM and GCCM are helpful for improving performance and reducing complexity, and our AFBO asymmetrically built with GCCM and OCCM can achieve better performance and complexity trade-off. Note that our model-agnostic

AFBO can be flexibly integrated with existing Transformer-like architectures, achieving better performance at low cost of complexity. The comparison of original FFN with our AFBO is illustrated in Fig. 1. To evaluate our AFBO, experiments are conducted on various vision tasks (i.e., image classification on ImageNet-1K (Krizhevsky et al., 2017) and its out-of-distribution variants (Hendrycks et al., 2021b; Hendrycks & Dietterich, 2019; Hendrycks et al., 2021a; Recht et al., 2019), object detection and instance segmentation on MS COCO (Lin et al., 2014)) with twenty ViT models (e.g., DeiT (Touvron et al., 2021a), Swin Transformer (Liu et al., 2021) and PoolFormer (Yu et al., 2022)). The contributions of this work can be summarized as follows:

(1) This paper proposes a novel Asymmetric Factorized Bilinear Operation (AFBO) as an effective alternative of FFN for ViTs. Particularly, our AFBO efficiently explores rich statistics of token features and shows great potential to achieve better performance and complexity trade-off for ViTs.

(2) To this end, our AFBO presents a spatial-channel factorized bilinear operation to efficiently model second-order statistics of token features and introduces two structured-sparsity channel mappings to reduce model complexity (i.e., parameters and FLOPs) while guaranteeing performance.

(3) Our AFBO is model-agnostic and can be flexibly integrated with existing Transformer-like architectures. Extensive experiments on various tasks by using different ViTs show our AFBO is superior to its counterparts, while improving existing ViTs in terms of generalization and robustness.

## 2 RELATED WORK

Building upon the remarkable success of transformer architecture in natural language processing (Vaswani et al., 2017), Vision Transformer (ViT) (Dosovitskiy et al., 2021) is initially introduced to extend transformer models to vision tasks (Krizhevsky et al., 2017; Fang et al., 2021; Strudel et al., 2021; Neimark et al., 2021). Subsequently, a lot of ViT variants (Wang et al., 2022; Liu et al., 2021; Vaswani et al., 2021; Ding et al., 2022; Huang et al., 2023; Ho et al., 2019; Lu et al., 2021) have been studied to improve ViT, especially for its self-attention mechanism. Most of these methods aim to reduce computational complexity of self-attention since its complexity is quadratic to token numbers. Among them, some works (Liu et al., 2021; Ding et al., 2022) focus on developing a local self-attention mechanism and its shifted/haloed version to add the interaction across different local windows. Besides, SOFT (Lu et al., 2021) replaces the dot-product similarity by proposing a softmax-free transformer with linear space and time complexity. Additionally, incorporation of self-attention with convolution as a hybrid backbone has been studied for enhancing locality of ViTs (Guo et al., 2022a; Wu et al., 2021; Dai et al., 2021; d'Ascoli et al., 2021; Li et al., 2023). There also exist some works (Yu et al., 2022) to challenge necessity of self-attention, and they propose some substituting operations (e.g., pooling and identity mapping) for attention while achieving comparable performance. Distinguished from the aforementioned works, our AFBO focuses on improving FFN module of ViT for achieving better efficiency and effectiveness trade-off.

Some recent works (Fang et al., 2024; Xu et al., 2024; Sridhar et al., 2023; Li et al., 2021) have been studied to improve channel mixing module (i.e., FFN) of ViT. For example, EVA-02 (Fang et al., 2024) extends GLU (Shazeer, 2020) to ViT. It employs a two-branch structure to replace FFN, which ensures computational efficiency by reducing the hidden dimension. Besides, some works (Xu et al., 2024; Sridhar et al., 2023) focus on reducing the computational complexity of FFN by developing some slimming schemes. Additionally, some studies (Li et al., 2021; Cao et al., 2023; Guo et al., 2022a) enhance the performance of FFN by introducing extra modules (e.g., channel attention (Hu et al., 2020; Wang et al., 2020) or DW convolution). TransNext (Shi, 2024) enhances FFN by combining SwiGLU (Shazeer, 2020; Fang et al., 2024) with DW convolution. Unlike these methods, our AFBO attempts to explore rich statistics of token features to improve FFN module of ViT, which achieves better performance and complexity trade-off than existing counterparts (refer to comparisons in Table 3).

## 3 PROPOSED METHOD

In this section, we first briefly revisit the original FFN for ViT, and describe details of our Asymmetric Factorized Bilinear Operation (AFBO). Finally, three AFBO variants are introduced and we compare them in terms of performance and complexity for model selection.

### 3.1 REVISITING FFN FOR VIT

As shown in Fig. 1 (a), the original FFN involved of two fully-connected (linear) layers with an activation function is used to perform channel mixing in transformer block. Let $\mathbf{X} \in \mathbb{R}^{d \times N}$ be $d$-dimensional features of $N$ tokens, the output of FFN (i.e., $\mathbf{Y} \in \mathbb{R}^{d \times N}$) can be written as

$$\mathbf{Y} = \otimes_{1 \times 1}^{\{\mathbf{W}_2, \mathbf{b}_2\}}\big(\sigma(\mathbf{Z})\big), \ \mathbf{Z} = \otimes_{1 \times 1}^{\{\mathbf{W}_1, \mathbf{b}_1\}}(\mathbf{X}), \tag{1}$$

where $\otimes_{1 \times 1}$ indicates point-wise convolution with kernel size of $1 \times 1$. $\{\mathbf{W}_1 \in \mathbb{R}^{d \times Md}, \mathbf{b}_1 \in \mathbb{R}^{Md}\}$ and $\{\mathbf{W}_2 \in \mathbb{R}^{Md \times d}, \mathbf{b}_2 \in \mathbb{R}^d\}$ respectively are weight parameters of two point-wise convolutions (*w.r.t* fully-connected layers), and $M$ is the expansion ratio of hidden dimension. $\mathbf{Z} \in \mathbb{R}^{Md \times N}$ means the intermediate features, and $\sigma(\cdot)$ is the activation function (e.g., GeLU (Hendrycks & Gimpel, 2016)). From Eq. (1), one can see that FFN performs feature learning through two linear projections, which do not fully consider the rich information lying in token features. A naive scheme to enhance performance of ViTs is to increase the hidden dimension of FFN, but it also brings more computational cost. By considering feature learning ability heavily influences performance, reasonable exploration of information lying in token features is a potential solution to improve ViT.

### 3.2 ASYMMETRIC FACTORIZED BILINEAR OPERATION (AFBO) FOR VIT

#### 3.2.1 SPATIAL-CHANNEL FACTORIZED BILINEAR OPERATION

Previous works (Lin et al., 2015; Wang et al., 2021a) show that the appropriate (global) modeling of second-order statistics lying in features can effectively improve deep CNNs. Inspired by these works, we incorporate idea of second-order modeling into FFN of ViT by modifying Eq. (1) as

$$\mathbf{Y} = \mathbf{W}_2^T \sigma(\mathbf{Z}) + \mathbf{b}_2, \ \mathbf{Z} = \widetilde{\mathbf{W}}_1^T (\mathbf{X}^T \mathbf{X}) + \mathbf{b}_1, \tag{2}$$

where $\mathbf{X}^T \mathbf{X}$ indicates the outer product of features $\mathbf{X}$, capturing second-order statistics of $\mathbf{X}$. $T$ indicates the transposition operation. However, such strategy in Eq. (2) suffers from two issues: (1) $\mathbf{X}^T \mathbf{X}$ generates a global representation, neglecting local information of each token; (2) representation size of $\mathbf{X}^T \mathbf{X}$ is $d \times d$, leading to a heavy computational burden, i.e., $\widetilde{\mathbf{W}}_1 \in \mathbb{R}^{d^2 \times Md}$.

To address above issues, our AFBO approximates computation of second-order statistics via a rank-1 decomposition, as suggested in (Gao et al., 2016; Li et al., 2017; Mnih & Hinton, 2007). As such, our AFBO replaces the first linear projection in FFN by a bilinear operation:

$$\mathbf{Y} = \otimes_{1 \times 1}^{\{\mathbf{W}_3, \mathbf{b}_3\}}\big(\sigma(\widehat{\mathbf{Z}})\big), \ \widehat{\mathbf{Z}} = [\widehat{\otimes}_{K \times K}^{\{\widehat{\mathbf{W}}_1, \widehat{\mathbf{b}}_1\}}(\mathbf{X})] \odot [\widehat{\otimes}_{K \times K}^{\{\widehat{\mathbf{W}}_2, \widehat{\mathbf{b}}_2\}}(\mathbf{X})], \tag{3}$$

where $\{\widehat{\mathbf{W}}_1 \in \mathbb{R}^{d \times Md \times K \times K}, \widehat{\mathbf{b}}_1 \in \mathbb{R}^{Md}\}$ and $\{\widehat{\mathbf{W}}_2 \in \mathbb{R}^{d \times Md \times K \times K}, \widehat{\mathbf{b}}_2 \in \mathbb{R}^{Md}\}$ are weight parameters of two convolutions, respectively. And $\widehat{\otimes}_{K \times K}$ indicates convolution operation with kernel size of $K \times K$. $\{\mathbf{W}_3 \in \mathbb{R}^{Md \times d}, \mathbf{b}_3 \in \mathbb{R}^d\}$ are weight parameters of point-wise convolution of $\otimes_{1 \times 1}$, and $\odot$ indicates the dot product.

For computational efficiency, we further present a spatial-channel factorized bilinear operation (SCFBO), which decomposes $\widehat{\otimes}_{K \times K}$ into a spatial modeling operation $\tilde{\otimes}_{K \times K}$ and a channel mapping (i.e., point-wise convolution $\otimes_{1 \times 1}$). According to Eq. (3), our SCFBO is formulated as

$$\mathbf{Y} = \otimes_{1 \times 1}^{\{\mathbf{W}_3, \mathbf{b}_3\}}\big(\sigma(\widehat{\mathbf{Z}})\big),$$
$$\widehat{\mathbf{Z}} = [\tilde{\otimes}_{K \times K}^{\{\tilde{\mathbf{W}}_1^s, \tilde{b}_1^s\}} \otimes_{1 \times 1}^{\{\mathbf{W}_1^c, \mathbf{b}_1^c\}}(\mathbf{X})] \odot [\tilde{\otimes}_{K \times K}^{\{\tilde{\mathbf{W}}_2^s, \tilde{b}_2^s\}} \otimes_{1 \times 1}^{\{\mathbf{W}_2^c, \mathbf{b}_2^c\}}(\mathbf{X})], \tag{4}$$

where $\{\mathbf{W}_\cdot^c \in \mathbb{R}^{d \times Md}, \mathbf{b}_\cdot^c \in \mathbb{R}^{Md}\}$ and $\{\tilde{\mathbf{W}}_\cdot^s \in \mathbb{R}^{K \times K}, \tilde{b}_\cdot^s \in \mathbb{R}\}$ are parameters of $\otimes_{1 \times 1}$ and $\tilde{\otimes}_{K \times K}$, respectively. Particularly, spatial modeling $\tilde{\otimes}_{K \times K}$ can be achieved by a DW convolution, local pooling or spatial attention (Woo et al., 2018) with kernel size of $K \times K$. As such, our AFBO with SCFBO (4) in Fig. 1 (b) has a two-branch structure with spatial modeling and channel mapping.

As for Eq. (4), bilinear representation $\widehat{\mathbf{Z}}$ is followed by an activation function $\sigma$ (e.g., SiLU (Elfwing et al., 2018) and GeLU (Hendrycks & Gimpel, 2016)). However, such bilinear operation potentially hurts the performance of such an activation function. Specifically, for a special case of parameter-shared branch in Eq. (4), $\widehat{\mathbf{Z}} = f(\mathbf{X})^2$ and $f$ indicates spatial-channel factorized mapping. Meanwhile, both outputs of SiLU and GeLU are power functions of inputs, which are more sensitive to

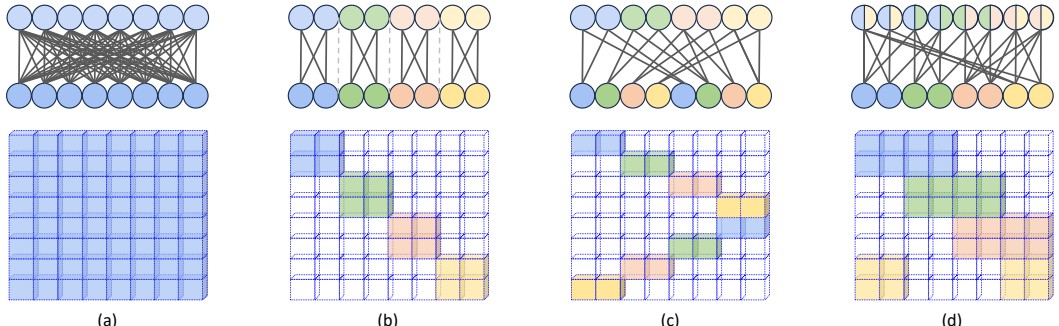

Figure 2: Diagram of mapping matrices for various channel mapping strategies. (a) Fully-connected layer with dense parameters. (b) Block-wise channel mapping with block-diagonal sparsity constraint ($G = 4$). (c) Our Grouped Cross Channel Mapping (GCCM) ($G = 4$). (d) Our Overlapped Cycle Channel Mapping (OCCM) ($G = 4$).

probable noise of $f(\mathbf{X})^2$ than one of $f(\mathbf{X})$, potentially decreasing the final performance. Therefore, as shown in Fig. 1, we switch dot product with activation function for our SCFBO in practice. More experimental comparisons can refer to Table 6.

### 3.2.2 STRUCTURED-SPARSITY CHANNEL MAPPING

By considering hidden dimension ($Md$) plays a crucial role in performance of ViT (Noshad et al., 2019; Sridhar et al., 2023), we are likely to adopt a large expansion ratio $M$ (e.g., $M = 4$) in channel mapping (i.e., point-wise convolution $\otimes_{1\times1}$ with parameters of $\{\mathbf{W}^c_. \in \mathbb{R}^{d \times Md}, \mathbf{b}^c_. \in \mathbb{R}^{Md}\}$) to guarantee performance, which subsequently leads to high computational complexity. Therefore, we further present two structured-sparsity channel mapping strategies to decrease computational complexity of SCFBO while guaranteeing performance. Given $\mathbf{x}_i$ be features of $i$-th token in $\mathbf{X}$, channel mapping performs a linear projection with parameters of $\{\mathbf{W}^c_. \in \mathbb{R}^{d \times Md}, \mathbf{b}^c_. \in \mathbb{R}^{Md}\}$ as $\mathbf{z}_i = \mathbf{W}^c_. \mathbf{x}_i + \mathbf{b}^c_.$. As illustrated in Fig. 2 (a), $\mathbf{W}^c_.$ is generally a fully-connected layer with $M \times d^2$ parameters. As a commonly used strategy, group channel mapping in Fig. 2 (b) has been studied to reduce computational complexity of deep CNNs (Chollet, 2017; Krizhevsky et al., 2017; Sandler et al., 2018), where $\mathbf{W}^c_.$ can be regarded as incorporation of a block-diagonal sparsity constraint:

$$\mathbf{W}^c_{.\,\mathrm{bw}} = \begin{bmatrix} \mathbf{W}^c_.(1) & \mathbf{0} & \cdots & \mathbf{0} \\ \mathbf{0} & \mathbf{W}^c_.(2) & \cdots & \mathbf{0} \\ \vdots & \vdots & \ddots & \vdots \\ \mathbf{0} & \mathbf{0} & \cdots & \mathbf{W}^c_.(G) \end{bmatrix}, \tag{5}$$

where $\mathbf{W}^c_.(i) \in \mathbb{R}^{Md/G \times d/G}$ and $G$ indicates number of group. Clearly, computational complexity of block-wise channel mapping in Eq. (5) is $1/G$ of the original $\mathbf{W}^c_.$. Although model efficiency, block-wise channel mapping suffers from inferior performance due to the absence of information interaction among different groups (refer to comparison in Table 1).

To guarantee both efficiency and effectiveness, we present two structured-sparsity channel mapping strategies, namely Grouped Cross Channel Mapping (GCCM) and Overlapped Cycle Channel Mapping (OCCM). As shown in Fig 2 (c), our GCCM divides the inputs and outputs into $G$ and $2G$ non-overlapped groups, respectively. Then, GCCM adopts a parameter-shared linear mapping for $i$-th and $i+G$-th groups ($i \in [1, G]$) of outputs by using different groups of inputs. As such, different groups of inputs are fed into a same linear mapping to realize information interaction between groups. Particularly, mapping matrix of our GCCM can be written as

$$\mathbf{W}^c_{.\,\mathrm{gc}} = \begin{bmatrix} \mathbf{W}^c_.(1) & \mathbf{0} & \cdots & \mathbf{0} & \cdots & \mathbf{0} & \cdots & \mathbf{W}^c_.(G) \\ \mathbf{0} & \mathbf{W}^c_.(2) & \cdots & \vdots & \vdots & \vdots & \ddots & \vdots \\ \vdots & \vdots & \ddots & \vdots & \mathbf{0} & \mathbf{W}^c_.(2) & \cdots & \vdots \\ \mathbf{0} & \mathbf{0} & \cdots & \mathbf{W}^c_.(G) & \mathbf{W}^c_.(1) & \mathbf{0} & \cdots & \mathbf{0} \end{bmatrix}^T, \tag{6}$$

Table 1: Comparisons of various channel mixers in terms of complexity, performance, and inference latency. Particularly, the parameter numbers in dashed boxes and the results of Top-1 accuracy are reported with instantiation of DeiT-T (Touvron et al., 2021a) on ImageNet-1K.

| Method | Parameters (M) | | FLOPs (G) | | Top-1 (%) | Latency (ms) |
|---|---|---|---|---|---|---|
| FFN | $\mathcal{O}(2Md^2)$ | 5.7 | $\mathcal{O}(4Md^2)$ | 1.3 | 72.2 | 11.4 |
| GLU ($M = \frac{8}{3}$) | $\mathcal{O}(3Md^2)$ | 5.7 | $\mathcal{O}(6Md^2)$ | 1.3 | 72.6 | 12.9 |
| Group FFN ($G = 2$) | $\mathcal{O}(\frac{2Md^2}{G} + Md^2)$ | 5.7 | $\mathcal{O}(\frac{4Md^2}{G} + 2Md^2)$ | 1.3 | 70.8 | 13.1 |
| Group FFN ($G = 4$) | $\mathcal{O}(\frac{2Md^2}{G^2} + Md^2)$ | 5.0 | $\mathcal{O}(\frac{4Md^2}{G} + 2Md^2)$ | 1.1 | 69.8 | 15.5 |
| SCFBO-GC | $\mathcal{O}(\frac{2Md^2}{G^2} + Md^2)$ | 5.1 | $\mathcal{O}(\frac{4Md^2}{G^2} + 2Md^2)$ | 1.1 | 72.2 | 12.7 |
| SCFBO-OC | $\mathcal{O}(\frac{2M(G-1)d^2}{G} + Md^2)$ | 6.7 | $\mathcal{O}(\frac{4M(G-1)d^2}{G} + 2Md^2)$ | 1.5 | 75.6 | 14.5 |
| AFBO | $\mathcal{O}(\frac{Md^2}{G_1^2} + \frac{M(G_2-1)d^2}{G_2} + Md^2)$ | 6.0 | $\mathcal{O}(\frac{2Md^2}{G_1} + \frac{2M(G_2-1)d^2}{G_2} + 2Md^2)$ | 1.3 | 74.6 | 13.2 |

where $\mathbf{W}^c_{\cdot}(i) \in \mathbb{R}^{Md/2G \times d/G}$, and our GCCM in Eq. (6) is only $1/2G$ of the original $\mathbf{W}^c_{\cdot}$ in term of computational complexity. Furthermore, as illustrated in Fig. 2 (d), our OCCM proposes to use overlapped inputs to realize information interaction between groups in a cycle manner:

$$
\mathbf{W}^c_{\underset{oc}{\cdot}} = \begin{bmatrix}
[\mathbf{W}^c_{\cdot L}(1), & \mathbf{W}^c_{\cdot R}(1)] & 0 & \cdots & 0 \\
0 & [\mathbf{W}^c_{\cdot L}(2), & \mathbf{W}^c_{\cdot R}(2)] & \cdots & 0 \\
\vdots & \vdots & \ddots & \ddots & \vdots \\
0 & 0 & \cdots & [\mathbf{W}^c_{\cdot L}(G-1), & \mathbf{W}^c_{\cdot R}(G-1)] \\
\mathbf{W}^c_{\cdot R}(G)] & 0 & \cdots & 0 & [\mathbf{W}^c_{\cdot L}(G)
\end{bmatrix}, \tag{7}
$$

where our OCCM sets the overlapped range by half of dimension of grouped features. $[\mathbf{W}^c_{\cdot L}(g) \in \mathbb{R}^{Md/G \times d/G}, \mathbf{W}^c_{\cdot R}(g) \in \mathbb{R}^{Md/G \times d/G}]$ are parameters of $g$-th group for channel mapping, whose numbers are $2/G$ of those in the original $\mathbf{W}^c_{\cdot}$. From Eq. (6) and Eq. (7) we can see that our GCCM and OCCM can efficiently perform information interaction between groups by introducing appropriate structured-sparsity constraints on channel mapping. Note that our AFBO can control the trade-off between performance and complexity by adopting various group numbers and overlap ratios.

### 3.3 AFBO VARIANTS

Based on the proposed GCCM and OCCM, we can implement SCFBO in Eq. (4) in different ways, resulting in three AFBO variants, i.e., SCFBO-GC, SCFBO-OC, and asymmetric SCFBO (AFBO).

**SCFBO-GC** We construct SCFBO-GC by realizing channel mappings of two branches lying in SCFBO with GCCM (6), which has parameter number of $\mathcal{O}(\frac{2Md^2}{G^2} + Md^2)$ and computational cost (FLOPs) of $\mathcal{O}(\frac{4Md^2}{G^2} + 2Md^2)$ with group number of $G$.

**SCFBO-OC** When channel mappings of two branches lying in SCFBO are replaced by OCCM (7), we construct SCFBO-OC with parameter number of $\mathcal{O}(\frac{2M(G-1)d^2}{G} + Md^2)$ and computational cost (FLOPs) of $\mathcal{O}(\frac{4M(G-1)d^2}{G} + 2Md^2)$, where $G$ is group number of OCCM.

**Asymmetric SCFBO (AFBO)** As a hybrid scheme, we replace left and right branches of SCFBO with OCCM (5) and GCCM (6), respectively. It leads to an asymmetric SCFBO, i.e., AFBO. Specifically, parameter number of AFBO is $\mathcal{O}(\frac{Md^2}{G_1^2} + \frac{M(G_2-1)d^2}{G_2} + Md^2)$, while computational cost is $\mathcal{O}(\frac{2Md^2}{G_1} + \frac{2M(G_2-1)d^2}{G_2} + 2Md^2)$, where $G_1$ and $G_2$ are group number of GCCM and OCCM.

**Model Selection via Complexity Analysis** To further analyze performance and complexity of three AFBO variants, we compare them with the original FFN, GLU (Shazeer, 2020), and Group FFN in Eq. (5). As listed in Table 1, our AFBO variants have similar or lower computational complexity than their counterparts from a theoretical perspective. By instantiating our AFBO variants with $G_1 = 2$ and $G_2 = 4$ with DeiT-T (Touvron et al., 2021a), our SCFBO-GC achieves similar performance with FFN but has fewer parameters and FLOPs. SCFBO-OC improves FFN by 3.4% and brings extra model complexity. Therefore, our AFBO combines with GCCM and OCCM to achieve performance and complexity trade-off, which is used throughout all experiments. Finally, our AFBO respectively brings 3.8% and 2.0% gains over Group FFN and GLU, while having comparable model complexity.

# 4 EXPERIMENTS

In this section, we first describe implementation details of our AFBO, and make comparison on ImageNet-1K (Krizhevsky et al., 2017) and its variants, i.e., ImageNet-C (Hendrycks & Dietterich, 2019), ImageNet-A (Hendrycks et al., 2021b), ImageNet-R (Hendrycks et al., 2021a) and ImageNet-Sketch (Recht et al., 2019). Besides, we transfer our models to object detection and instance segmentation on MS COCO (Lin et al., 2014). Finally, we conduct ablation studies on ImageNet-1K.

## 4.1 IMPLEMENTATION DETAILS

As a plug-in module, we apply our AFBO to various ViTs and MLP-Mixer architectures by replacing the original FFN modules, including DeiT (Touvron et al., 2021a), Swin ViT (Liu et al., 2021), PoolFormer (Yu et al., 2022), LVT (Yang et al., 2022), PVTv2 (Wang et al., 2022), CycleMLP (Chen et al., 2023), HireMLP-Tiny (Guo et al., 2022b), GC ViT (Hatamizadeh et al., 2023) and VisionL-LaMA (Chu et al., 2024). To match the original FFN in terms of model complexity, we set $G_1$ and $G_2$ to 2 and 3 for AFBO, respectively. For spatial modeling, we use $3\times3$ DW convolution as a default setting. To train models on ImageNet-1K, we adopt exactly same strategies as the original works with $224 \times 224$ inputs. For evaluation on object detection and instance segmentation, we adopt Mask R-CNN (He et al., 2017) and RetinaNet (Lin et al., 2020) as baseline detectors, where Poolformer-S12 (Yu et al., 2022) and Swin-T (Liu et al., 2021) along with FPN (Lu et al., 2020) are used as backbone models. All detectors are implemented using MMDetection toolkit (Chen et al., 2019) with the default settings. Specifically, the shorter side of input images is resized to 800, and all the models are optimized using SGD with weight decay of 1e-4, momentum of 0.9 and mini-batch size of 16. The learning rate is initialized to 0.01 and is decreased by a factor of 10 after 8 and 11 epochs, respectively. All programs are implemented by PyTorch (Paszke et al., 2019) and run on a server with 8 A6000 GPUs. The source code is available at https://github.com/XavierHeart/AFBO.

## 4.2 IMAGE CLASSIFICATION ON IMAGENET-1K

We first validate the effectiveness of our AFBO by comparing state-of-the-art (SOTA) models and its counterparts on ImageNet-1K. Besides, we directly adopt the trained models to ImageNet-C (Hendrycks & Dietterich, 2019), ImageNet-A (Hendrycks et al., 2021b), ImageNet-R (Hendrycks et al., 2021a) and ImageNet-Sketch (Recht et al., 2019) to verify the robustness of our AFBO.

### 4.2.1 COMPARISON WITH SOTA

To verify generalization of our AFBO, we compare 20 SOTA models with and without AFBO. As shown in Table 2, our AFBO enhances the performance of all models on both IN-1K and 4 out-of-distribution (OOD) variants, while having less or comparable model complexity (i.e., parameters and FLOPs). Specifically, for tiny ViT models, AFBO respectively achieves 1.3% and 2.4% gains over LVT and DeiT-T on IN-1K, while bringing clear improvement on four OOD variants. For small ViT models, AFBO obtains 0.5%~1.9% and 0.6%~5.6% gains on IN-1K and four OOD variants, respectively. Notably, AFBO achieves 0.5% gains over the recently proposed GC ViT-XT and Pyramid VisionLLaMA-S on IN-1K, and brings more than 1.5% gains on IN-A. Besides, AFBO can also improve MLP-Mixer models (i.e., CycleMLP and HireMLP) with comparable model complexity. For middle and large ViT models, AFBO respectively brings 0.7% and 0.1% gains for DeiT-B and Swin-B on IN-1K, but has less model complexity. In particular, AFBO still brings clear improvement (0.6%~3.7%) for medium and large models on four OOD variants. In terms of model latency, AFBO brings extra affordable inference time over the original models. These results above clearly demonstrate that AFBO can help existing ViTs achieve better performance and complexity trade-off, which provides a promising solution to improve generalization and robustness of ViTs.

### 4.2.2 COMPARISON WITH COUNTERPARTS

To further evaluate the effectiveness of our AFBO, we compare with several counterparts, including SwiGLU (Fang et al., 2024), ConvNeXt block (Liu et al., 2022), IMLP (Xu et al., 2024), SCHEME (Sridhar et al., 2023) and ConvGLU (Shi, 2024), where all experiments are conducted on IN-1K and IN-R by using DeiT (Touvron et al., 2021a), Poolformer (Yu et al., 2022) and Swin-T (Liu et al., 2021) as backbone models. Since source code is unavailable, we duplicate the results of

Table 2: Comparisons with various widely used vision models on ImageNet-1K (IN-1K) and four variants, including ImageNet-C (IN-C), ImageNet-A (IN-A), ImageNet-Robustness (IN-R) and ImageNet-Sketch (IN-SK). For comparison, the original MLP modules for all baseline models are replaced by our proposed AFBO module. Among OOD datasets, IN-C calculates the mean corruption error (mCE) as metric, where the smaller mCE means the better robustness of the models under corruptions. All other benchmarks use Top-1 accuracy as the metric if no special illustration.

| Method | Params. (M) | FLOPs (G) | Latency (ms) | IN-1K ($\uparrow\%$) | IN-C ($\downarrow\%$) | IN-A ($\uparrow\%$) | IN-R ($\uparrow\%$) | IN-SK ($\uparrow\%$) |
|---|---|---|---|---|---|---|---|---|
| LVT (Yang et al., 2022) | 5.5 | 0.8 | 7.7 | 74.8 | 75.0 | 7.5 | 34.6 | 23.0 |
| + AFBO (ours) | -0.3 | +0.0 | 9.1 | $76.1_{(1.3)}$ | $73.3_{(1.7)}$ | $8.5_{(1.0)}$ | $35.1_{(0.5)}$ | $23.7_{(0.7)}$ |
| DeiT-T (Touvron et al., 2021a) | 5.7 | 1.3 | 11.4 | 72.2 | 71.1 | 7.3 | 32.6 | 20.2 |
| + AFBO (ours) | +0.3 | +0.0 | 13.2 | $74.6_{(2.4)}$ | $66.2_{(4.9)}$ | $8.3_{(1.0)}$ | $38.5_{(5.9)}$ | $23.1_{(2.9)}$ |
| PoolFormer-S12 (Yu et al., 2022) | 11.9 | 1.8 | 9.9 | 77.2 | 69.8 | 7.0 | 37.7 | 25.4 |
| + AFBO (ours) | +0.1 | +0.1 | 11.2 | $79.1_{(1.9)}$ | $65.0_{(4.8)}$ | $8.9_{(1.9)}$ | $42.5_{(4.8)}$ | $27.6_{(2.2)}$ |
| GC ViT-XXT (Hatamizadeh et al., 2023) | 12.0 | 2.1 | 11.1 | 79.9 | 72.9 | 19.0 | 41.9 | 29.2 |
| + AFBO (ours) | +0.3 | +0.2 | 12.5 | $81.2_{(1.3)}$ | $72.5_{(0.4)}$ | $22.4_{(3.4)}$ | $43.5_{(1.6)}$ | $31.6_{(1.4)}$ |
| PVTv2-B1 (Wang et al., 2022) | 14.0 | 2.1 | 11.5 | 78.7 | 65.1 | 14.6 | 41.7 | 28.9 |
| + AFBO (ours) | -0.1 | +0.2 | 13.2 | $80.1_{(1.4)}$ | $63.6_{(1.5)}$ | $16.7_{(2.1)}$ | $44.1_{(2.4)}$ | $31.1_{(2.2)}$ |
| CycleMLP-B1 (Chen et al., 2023) | 15.0 | 2.1 | 12.1 | 78.9 | 64.5 | 11.6 | 41.6 | 29.1 |
| + AFBO (ours) | -0.2 | +0.1 | 13.4 | $80.0_{(1.1)}$ | $62.3_{(2.2)}$ | $14.2_{(2.6)}$ | $44.3_{(2.7)}$ | $30.4_{(1.3)}$ |
| HireMLP-Tiny (Guo et al., 2022b) | 17.0 | 2.1 | 12.8 | 78.9 | 65.3 | 12.8 | 41.5 | 29.0 |
| + AFBO (ours) | -0.2 | +0.1 | 15.1 | $80.2_{(1.3)}$ | $62.9_{(2.4)}$ | $15.6_{(2.8)}$ | $43.8_{(2.3)}$ | $30.5_{(1.5)}$ |
| GC ViT-XT (Hatamizadeh et al., 2023) | 20.0 | 2.6 | 13.2 | 82.0 | 75.3 | 26.7 | 44.3 | 32.0 |
| + AFBO (ours) | +0.4 | +0.2 | 14.9 | $82.5_{(0.5)}$ | $74.8_{(0.5)}$ | $29.2_{(2.5)}$ | $44.9_{(0.6)}$ | $32.6_{(0.6)}$ |
| Pyramid-VisionLLaMA-S (Chu et al., 2024) | 22.0 | 2.6 | 14.8 | 81.6 | 58.1 | 23.5 | 41.8 | 28.9 |
| + AFBO (ours) | +0.1 | +0.1 | 16.6 | $82.1_{(0.5)}$ | $56.1_{(2.0)}$ | $25.3_{(1.8)}$ | $45.3_{(3.5)}$ | $32.9_{(3.0)}$ |
| Poolformer-S24 (Yu et al., 2022) | 21.4 | 3.4 | 14.3 | 80.3 | 62.2 | 14.5 | 41.4 | 28.9 |
| + AFBO (ours) | +0.6 | +0.3 | 16.4 | $81.5_{(1.2)}$ | $57.8_{(4.4)}$ | $18.0_{(3.5)}$ | $44.5_{(3.1)}$ | $30.8_{(1.9)}$ |
| DeiT-S (Touvron et al., 2021a) | 22.0 | 4.6 | 12.9 | 79.8 | 54.6 | 19.8 | 41.9 | 29.4 |
| + AFBO (ours) | -0.3 | -0.1 | 15.2 | $81.1_{(1.3)}$ | $53.3_{(1.3)}$ | $20.8_{(1.0)}$ | $45.3_{(3.4)}$ | $31.9_{(2.5)}$ |
| Swin-T (Liu et al., 2021) | 28.0 | 4.5 | 12.6 | 81.2 | 62.0 | 21.7 | 41.3 | 29.0 |
| + AFBO (ours) | -0.5 | -0.1 | 15.5 | $82.1_{(0.9)}$ | $56.4_{(5.6)}$ | $26.0_{(4.3)}$ | $45.8_{(4.5)}$ | $31.7_{(2.7)}$ |
| PoolFormer-S36 (Yu et al., 2022) | 30.9 | 5.0 | 16.6 | 81.4 | 60.0 | 18.5 | 42.1 | 30.2 |
| + AFBO (ours) | +0.8 | -0.5 | 18.8 | $81.9_{(0.5)}$ | $57.1_{(2.9)}$ | $21.8_{(3.3)}$ | $43.8_{(1.7)}$ | $31.6_{(1.4)}$ |
| Swin-S (Liu et al., 2021) | 50.0 | 8.7 | 25.2 | 83.0 | 54.9 | 32.9 | 44.9 | 32.0 |
| + AFBO (ours) | -1.9 | -0.2 | 29.9 | $83.3_{(0.3)}$ | $52.1_{(2.8)}$ | $33.3_{(0.4)}$ | $47.4_{(2.5)}$ | $34.5_{(2.5)}$ |
| Pyramid-VisionLLaMA-B (Chu et al., 2024) | 56.0 | 9.0 | 31.3 | 83.2 | 52.1 | 33.5 | 46.0 | 33.5 |
| + AFBO (ours) | +0.4 | +0.2 | 32.9 | $83.5_{(0.2)}$ | $49.5_{(2.6)}$ | $35.2_{(1.7)}$ | $48.1_{(2.1)}$ | $35.8_{(2.3)}$ |
| PoolFormer-M36 (Yu et al., 2022) | 56.2 | 8.8 | 29.3 | 82.1 | 58.3 | 23.8 | 43.3 | 30.6 |
| + AFBO (ours) | +1.0 | +0.9 | 33.6 | $82.5_{(0.4)}$ | $54.6_{(3.7)}$ | $27.5_{(3.7)}$ | $45.2_{(1.9)}$ | $32.1_{(1.5)}$ |
| PoolFormer-M48 (Yu et al., 2022) | 73.5 | 11.6 | 31.6 | 82.5 | 54.9 | 25.7 | 43.7 | 30.9 |
| + AFBO (ours) | +1.4 | +1.2 | 35.7 | $83.0_{(0.5)}$ | $52.8_{(2.1)}$ | $32.0_{(3.7)}$ | $46.6_{(2.9)}$ | $32.0_{(1.1)}$ |
| DeiT-B (Touvron et al., 2021a) | 86.0 | 17.6 | 35.7 | 81.8 | 48.5 | 27.4 | 44.9 | 32.0 |
| + AFBO (ours) | -2.1 | -0.2 | 38.8 | $82.5_{(0.7)}$ | $45.3_{(3.2)}$ | $31.1_{(3.7)}$ | $47.4_{(2.5)}$ | $33.3_{(1.3)}$ |
| Swin-B (Liu et al., 2021) | 87.8 | 15.4 | 34.2 | 83.5 | 54.4 | 35.8 | 46.6 | 32.4 |
| + AFBO (ours) | -3.1 | -0.4 | 37.3 | $83.6_{(0.1)}$ | $51.5_{(2.9)}$ | $36.4_{(0.6)}$ | $49.0_{(2.4)}$ | $34.0_{(1.6)}$ |
| Pyramid-VisionLLaMA-L (Chu et al., 2024) | 99.0 | 18.0 | 40.1 | 83.6 | 54.4 | 38.7 | 47.8 | 34.6 |
| + AFBO (ours) | +0.5 | +0.2 | 42.9 | $83.7_{(0.1)}$ | $52.0_{(2.4)}$ | $40.2_{(1.5)}$ | $50.2_{(2.4)}$ | $36.2_{(1.6)}$ |

SCHEME (Sridhar et al., 2023) and IMLP (Xu et al., 2024) from the original papers for comparison. Then, we re-implement SwiGLU, ConvNeXt block and ConvGLU for the compared backbones using available source code. As shown in Table 3, IMLP has less model complexity but inferior performance. SCHEME is also difficult to balance performance and complexity. Particularly, our AFBO achieves higher accuracy than SCHEME-12, while having less model complexity. Compared to SwiGLU, ConvNeXt block and ConvGLU, AFBO has similar model complexity, while benefiting better generalization and robustness. Specifically, AFBO achieves 0.3%~2.0% and 0.4%~5.1% gains over SwiGLU, ConvGLU and ConvNeXt block on IN-1K and IN-R, respectively. Above results clearly show our AFBO can achieve better performance and complexity trade-off again.

### 4.3 OBJECT DETECTION AND INSTANCE SEGMENTATION ON COCO

Furthermore, we evaluate generalization ability of AFBO on object detection and instance segmentation tasks. As shown in Table 4, when RetinaNet is used as a detector, AFBO brings 1.7% and 0.6% gains over Poolformer-S12 and Swin-T in terms of AP, respectively. For Mask R-CNN,

Table 3: Comparisons with various counterparts on ImageNet-1K (IN-1K) and ImageNet-Robustness (IN-R), where the results of "FFN", "IMLP" and "SCHEME" are duplicated from the original works (Xu et al., 2024; Sridhar et al., 2023). We re-implement "ConvNext", "SwiGLU" and "ConvGLU" by using publicly available source code.

| Backbone | Method | Parameter | FLOPs | IN-1K | IN-R |
|---|---|---|---|---|---|
| DeiT-T (Touvron et al., 2021a) | FFN | 5.7 M | 1.3 G | 72.2% | 32.6% |
| | ConvNeXt Block (Liu et al., 2022) | 5.9 M | 1.3 G | 73.6% | 34.2% |
| | IMLP (Xu et al., 2024) | 5.0 M | 1.1 G | 72.6% | 33.5% |
| | SwiGLU (Fang et al., 2024) | 5.7 M | 1.3 G | 72.6% | 33.4% |
| | ConvGLU (Shi, 2024) | 5.9 M | 1.4 G | 74.2% | 37.5% |
| | **AFBO (ours)** | **6.0 M** | **1.3 G** | **74.6%** | **38.5%** |
| DeiT-S (Touvron et al., 2021a) | FFN | 22.0 M | 4.6 G | 79.8% | 41.9% |
| | IMLP (Xu et al., 2024) | 18.8 M | 3.9 G | 80.0% | N/A |
| | SwiGLU (Fang et al., 2024) | 22.0 M | 4.6 G | 80.4% | 42.5% |
| | ConvGLU (Shi, 2024) | 21.6 M | 4.5 G | 80.6% | 44.7% |
| | **AFBO (ours)** | **21.7 M** | **4.5 G** | **81.1%** | **45.3%** |
| Pool-S12 (Yu et al., 2022) | FFN | 11.9 M | 1.8 G | 77.2% | 37.7% |
| | IMLP (Xu et al., 2024) | 9.8 M | 1.5 G | 77.2% | N/A |
| | SCHEME-12 (Sridhar et al., 2023) | 16.7 M | 2.6 G | 78.5% | N/A |
| | SCHEME-44 (Sridhar et al., 2023) | 7.2 M | 1.0 G | 73.0% | N/A |
| | ConvGLU (Shi, 2024) | 12.0 M | 1.9 G | 78.1% | 41.7% |
| | **AFBO (ours)** | **12.0 M** | **1.9 G** | **79.1%** | **42.5%** |
| Pool-S24 (Yu et al., 2022) | FFN | 21.4 M | 3.4 G | 80.3% | 41.4% |
| | IMLP (Xu et al., 2024) | 17.2 M | 2.7 G | 80.7% | N/A |
| | SCHEME-12 (Sridhar et al., 2023) | 30.8 M | 4.9 G | 80.5% | N/A |
| | ConvGLU (Shi, 2024) | 22.0 M | 3.5 G | 81.1% | 43.8% |
| | **AFBO (ours)** | **22.0 M** | **3.7 G** | **81.5%** | **44.5%** |
| Swin-T (Liu et al., 2021) | FFN | 28.0 M | 4.5 G | 81.2% | 41.3% |
| | IMLP (Xu et al., 2024) | 24.3 M | 3.9 G | 81.5% | N/A |
| | SCHEME-44 (Sridhar et al., 2023) | 19.7 M | 3.1 G | 79.6% | N/A |
| | SCHEME-12 (Sridhar et al., 2023) | 36.9 M | 5.9 G | 81.7% | N/A |
| | ConvGLU (Shi, 2024) | 28.5 M | 4.7 G | 81.8% | 44.8% |
| | **AFBO (ours)** | **27.5 M** | **4.4 G** | **82.1%** | **45.8%** |

AFBO respectively brings 1.2% and 0.8% gains over Poolformer-S12 and Swin-T on object detection tasks, while achieving 1.2% and 1.3% improvement on instance segmentation, respectively. Besides, Poolformer-S12 and Swin-T with our AFBO are superior to other compared backbone models that share similar parameters. These results demonstrate that our AFBO can be well generalized to various tasks, e.g., object detection and instance segmentation.

## 4.4 ABLATION STUDIES ON IMAGENET-1K

In this subsection, we conduct ablation studies to assess the effects of core components and various configurations of our AFBO. All experiments are performed on IN-1K with the backbone of DeiT-T.

**Effect of Different Core Components.** Our AFBO involves two core components, i.e., spatial modeling (SM) and channel mapping (CM). To assess their effect, we construct various variants, including AFBO with only CM (AFBO-C), AFBO with only SM (AFBO-S), AFBO with CM followed by SM (AFBO-CS), and AFBO with SM followed by CM (AFBO-SC). The results are presented in Table 5, where we can see that both SM and CM modules improve DeiT-T. Besides, AFBO-C is superior to AFBO-S with fewer parameters, indicating that CM is more important than SM. AFBO-CS achieves further gains (0.9%~1.3%) by combining CM with the subsequent SM, meaning that SM and CM are complementary. Compared with AFBO-CS, AFBO-SC decreases model complexity, but achieves inferior accuracy. More analysis on SM and CM can refer to the appendix.

**Effect of Various Configurations.** Furthermore, we evaluate the effect of microcosmic configurations on our AFBO. Specifically, we analyze how position and type of activation function ($\sigma$) impact final performance. In our AFBO, $\sigma$ can be placed before or after the dot product operation, indicated by pre-act and post-act, respectively. For the type of activation function, we compare ReLU (Nair & Hinton, 2010), GELU and SiLU (Hendrycks & Gimpel, 2016). As listed in Table 6, post-act

Table 4: Results of object detection and instance segmentation on MS COCO, where all backbones are pretrained on ImageNet-1K and $1\times$ learning schedule is used. $AP^b$ and $AP^m$ denote bounding box $AP$ and mask $AP$, respectively. TP denotes Throughput, i.e., images per second.

| | Params. (M) | Flops (G) | TP (HZ) | $AP$ (%) | $AP_{50}$ (%) | $AP_{75}$ (%) | $AP_S$ (%) | $AP_M$ (%) | $AP_L$ (%) |
|---|---|---|---|---|---|---|---|---|---|
| | | | | RetinaNet $1\times$ | | | | | |
| ResNet-18 (He et al., 2016) | 21.3 | 173 | 29.4 | 31.8 | 49.6 | 33.6 | 16.3 | 34.3 | 43.2 |
| PoolFormer-S12 (Yu et al., 2022) | 21.7 | 189 | 18.5 | 36.2 | 56.2 | 38.2 | 20.8 | 39.1 | 48.0 |
| PoolFormer-S12+AFBO | 22.0 | 189 | 17.9 | $37.9_{1.7}$ | $57.2_{1.0}$ | $40.1_{1.9}$ | $21.3_{0.5}$ | $41.0_{0.9}$ | $50.2_{2.2}$ |
| PVT-Tiny (Wang et al., 2021b) | 23.0 | 201 | 23.5 | 36.7 | 56.9 | 38.9 | 22.6 | 38.8 | 50.0 |
| PVT-Small (Wang et al., 2021b) | 34.2 | 258 | 16.4 | 40.4 | 61.3 | 43.0 | 25.0 | 42.9 | 55.7 |
| ResNet-50 (He et al., 2016) | 37.7 | 220 | 26.3 | 36.3 | 55.3 | 38.6 | 19.3 | 40.0 | 48.8 |
| Swin-T (Liu et al., 2021) | 38.5 | 245 | 20.4 | 41.5 | 62.1 | 44.2 | 25.1 | 44.9 | 55.5 |
| Swin-T+AFBO | 38.0 | 245 | 19.6 | $42.1_{0.6}$ | $62.6_{0.5}$ | $44.9_{0.7}$ | $25.4_{0.3}$ | $45.5_{0.6}$ | $56.4_{0.9}$ |

| | Params. (M) | Flops (G) | TP (HZ) | $AP^b$ (%) | $AP_{50}^b$ (%) | $AP_{75}^b$ (%) | $AP^m$ (%) | $AP_{50}^m$ (%) | $AP_{75}^m$ (%) |
|---|---|---|---|---|---|---|---|---|---|
| | | | | Mask R-CNN $1\times$ | | | | | |
| ResNet-18 (He et al., 2016) | 31.2 | 198 | 19.6 | 34.0 | 54.0 | 36.7 | 31.2 | 51.0 | 32.7 |
| PoolFormer-S12 (Yu et al., 2022) | 31.6 | 207 | 13.0 | 37.3 | 59.0 | 40.1 | 34.6 | 55.8 | 36.9 |
| PoolFormer-S12+AFBO | 32.0 | 207 | 12.4 | $38.6_{1.2}$ | $60.2_{1.2}$ | $42.4_{2.3}$ | $35.8_{1.2}$ | $56.3_{0.5}$ | $38.2_{1.3}$ |
| PVT-Tiny (Wang et al., 2021b) | 32.9 | 225 | 14.9 | 36.7 | 59.2 | 39.3 | 35.1 | 56.7 | 37.3 |
| Twins-SVT-S | 44.0 | 228 | 14.4 | 42.7 | 65.6 | 46.7 | 39.6 | 62.5 | 42.6 |
| CMT-S (Guo et al., 2022a) | 44.5 | 249 | N/A | 44.6 | 66.8 | 48.9 | 40.7 | 63.9 | 43.4 |
| PVT-Small (Wang et al., 2021b) | 44.1 | 282 | 12.4 | 40.4 | 62.9 | 43.8 | 37.8 | 60.1 | 40.3 |
| ResNet50 (He et al., 2016) | 44.2 | 246 | 17.0 | 38.0 | 58.6 | 41.4 | 34.4 | 55.1 | 36.7 |
| Swin-T (Liu et al., 2021) | 48.0 | 267 | 14.3 | 43.7 | 66.6 | 47.7 | 39.8 | 63.3 | 42.7 |
| Swin-T+AFBO | 47.5 | 267 | 13.6 | $44.5_{0.8}$ | $67.2_{0.6}$ | $48.2_{0.5}$ | $41.1_{1.3}$ | $63.7_{0.4}$ | $42.9_{0.3}$ |

Table 5: Effect of core components on AFBO.

| Method | Description | Params. | Top-1 |
|---|---|---|---|
| Baseline | – | 5.7 M | 72.2% |
| AFBO-C | Only Channel Mapping | 5.7 M | 73.7% |
| AFBO-S | Only Spatial Modeling | 6.0 M | 73.3% |
| AFBO-CS | Channel Mapping followed by Spatial Modeling | 6.0 M | 74.6% |
| AFBO-SC | Spatial Modeling followed by Channel Mapping | 5.8 M | 73.4% |

Table 6: Effect of different configurations.

| | OCCM | GCCM | $\sigma$ | Top-1 | Latency |
|---|---|---|---|---|---|
| Pre-act | – | | SiLU | 73.5% | 13.2 ms |
| | – | | ReLU | 73.7% | 12.6 ms |
| Pre-act | | ✓ | SiLU | 74.2% | 13.2 ms |
| | ✓ | | SiLU | 74.6% | 13.2 ms |
| | ✓ | | GeLU | 74.6% | 13.1 ms |
| | | ✓ | ReLU | 73.7% | 12.6 ms |
| | ✓ | | ReLU | 74.2% | 12.7 ms |
| | ✓ | ✓ | SiLU | 74.3% | 13.1 ms |

strategy is clearly inferior to pre-act, whose reason is analyzed in Sec. 3.2.1. The similar philosophy is also observed in GLU (Shazeer, 2020). Besides, ReLU is inferior to SiLU and GELU in terms of accuracy while having slightly less model latency. Additionally, the combination of activation function with OCCM branch outperforms one of GCCM and both of them. It may be caused by the fact that OCCM has better feature learning ability than GCCM (i.e., SCFBO-OC performs better than SCFBO-GC in Table 1), and the dot product of two activation functions may hurt feature learning ability, leading an inferior performance. As a contrast, ReLU is more tolerated for dot product operation. In this work, SiLU activation with OCCM is used as the default setting.

## 5 CONCLUSION

This paper made an attempt to solve the paradox of performance and complexity trade-off in improving FFN of ViTs. To this end, we proposed an Asymmetric Factorized Bilinear Operation (AFBO), whose core idea is to explore rich statistics of token features to improve feature learning ability of Transformer in an efficient way. Specifically, our AFBO presents a spatial-channel factorized bilinear operation to model second-order statistics for feature learning, while two structured-sparsity channel mappings are developed to reduce computational complexity while guaranteeing performance. Extensive experiments on various tasks by using several ViT models clearly demonstrate our AFBO has a good ability to achieve performance and complexity trade-offs. In the future, we will adopt our AFBO to foundation models and other applications (e.g., semantic segmentation (Zhou et al., 2017) and visual question answering (Antol et al., 2015)).

ACKNOWLEDGMENTS

This work was supported in part by the National Natural Science Foundation of China under Grant 62276186, Grant 61925602, Grant 62222608, and Grant 62436002; in part by the National Science and Technology Major Project under Grant 2022ZD0116500; and in part by the CAAI-Huawei MindSpore Open Fund under Grant CAAIXSJLJJ-2022-010C; and in part by the Haihe Lab of ITAI under Grant 22HHXCJC00002.

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

## A    APPENDIX

Table 7: Results of AFBO with different groups numbers $G1$ and $G2$. $\uparrow$ indicates higher is better, and $\downarrow$ indicates lower is better.

| $G1$ | $G2$ | Param (M) | Flops (G) | Latency (ms) | IN-1K (%) | IN-A ($\uparrow$%) | IN-R ($\uparrow$%) | IN-SK ($\uparrow$%) | IN-C ($\downarrow$%) |
|---|---|---|---|---|---|---|---|---|---|
| 2 | 6 | 6.2 | 1.4 | 13.7 | 74.8 | 8.3 | 38.8 | 23.6 | 66.0 |
| 2 | 4 | 6.0 | 1.3 | 13.2 | 74.6 | 8.3 | 38.5 | 23.1 | 66.2 |
| 2 | 3 | 5.8 | 1.3 | 13.1 | 74.2 | 8.1 | 36.7 | 22.7 | 67.1 |
| 2 | 2 | 5.6 | 1.3 | 12.9 | 73.0 | 7.6 | 34.6 | 21.0 | 70.4 |
| 4 | 4 | 5.7 | 1.3 | 12.8 | 73.5 | 8.1 | 36.9 | 21.4 | 68.9 |

### A.1    EFFECT OF GROUP NUMBERS OF GCCM AND OCCM

In this subsection, we assess the effect of group numbers $G1$ and $G2$ by conducting experiments with a backbone of DeiT-T (Touvron et al., 2021a) on five datasets, including IN-1K, IN-A, IN-R, IN-SK, and IN-C. The results are given in Table 7, where we can see that larger numbers of $G2$ bring higher accuracies and more computational costs. In contrast, larger numbers of $G1$ decrease both accuracies and computational cost. Particularly, performances of varying group numbers $G1$ and $G2$ are consistent across four benchmarks. To balance efficiency and effectiveness, we respectively set $G1 = 2$ and $G2 = 3$ of GCCM and OCCM for LVT (Yang et al., 2022), Swin (Liu et al., 2021) and MLP-Mixers (Chen et al., 2023; Guo et al., 2022b) as the default settings. For the other remaining models, we set $G1 = 2$ and $G2 = 4$ for GCCM and OCCM, respectively.

### A.2    EFFECT OF DIFFERENT SPATIAL MODELING STRATEGIES

The spatial modeling (SM) of our AFBO can be achieved by various strategies, including parameter-free operations (i.e., spatial max-pooling and avg-pooling), spatial attention and DW convolution. To investigate the effect of different spatial modeling strategies on our AFBO, we compare them on ImageNet-1K using the DeiT-T backbone. As shown in Table 8, parameter-free operations, despite their computational efficiency, result in poor performance. We implement spatial attention as suggested in (Woo et al., 2018), which has fewer parameters than DW convolution but also achieves

Table 8: Comparison of different SM.

| Backbone | Method | Params. | Top-1 |
|----------|--------|---------|-------|
| DeiT-T | Max-pooling | 5.7 M | 66.4% |
| | Avg-pooling | 5.7 M | 50.0% |
| | Spatial Attention | 5.8 M | 71.0% |
| | DW Conv. | 6.0 M | 74.6% |

Table 9: Effect of kernel size on SM.

| Kernel Size | Params. | Top-1 |
|-------------|---------|-------|
| – | 5.7 M | 73.7% |
| 3 | 6.0 M | 74.6% |
| 5 | 6.2 M | 74.7% |
| 7 | 6.5 M | 74.9% |

Table 10: Comparison of DW convolution (Conv) with original convolution as spatial modeling of AFBO in terms of model complexity, where DeiT-T and Swin-T are used as backbones.

| Backbone | Method | Params. (M) | Flops (G) | Latency (ms) |
|----------|--------|-------------|-----------|--------------|
| DeiT-T | DW Conv (3×3) | 6.0 | 1.3 | 13.2 |
| DeiT-T | Conv (3×3) | 13.3 | 1.7 | 18.6 |
| Swin-T | DW Conv (3×3) | 27.5 | 4.4 | 15.5 |
| Swin-T | Conv (3×3) | 64.8 | 4.6 | 28.3 |

inferior performance. Therefore, we use DW convolution as the default spatial modeling strategy of our AFBO.

Furthermore, we explore the effect of different convolution kernel sizes on DW convolution. In Table 9, we compare results of DW convolution with different kernel sizes. We observe that larger kernel sizes lead to greater performance gains, but also increase the computational cost. Besides, a better balance between performance and efficiency is achieved when the kernel size is set to 3. Therefore, we set the kernel size of 3 as default setting of DW convolution in our AFBO.

Additionally, we compare DW convolution with original convolution for spatial modeling. As shown in Table 10, when DW convolution is replaced by the original convolution with a kernel size of 3x3, parameters increase more than 1x times for DeiT-T (Touvron et al., 2021a) and Swin-T (Liu et al., 2021) models. Meanwhile, model latency (inference time per image) increases by about 50%~85%. Therefore, DW convolution can significantly reduce the computational cost and guarantee the efficiency of the whole model, compared with the original convolution.

## A.3 COMPARISON OF GLU WITH LARGE EXPANSION RATIO OF HIDDEN DIMENSION

Since hidden dimension of FFN heavily affects performance of ViTs, we compare it with GLU by using an expansion ratio of $M = 4$ to further evaluate the effectiveness of our AFBO. Specifically, experiments are conducted on ImageNet-1K (IN-1K) with backbones of Poolformer-S12, Poolformer-S24, and Swin-T. As shown in Table 11, for an expansion ratio of $M = 4$, GLU suffers from much higher computational cost, while performance still lags behind our AFBO in terms of both generalization and robustness on IN-1K and IN-R. These results clearly show that our AFBO can achieve better performance and complexity trade-off for ViTs.

## A.4 COMPARISON WITH BLOCK-WISE CHANNEL MAPPING

In this subsection, we provide more comparisons of our proposed channel mapping strategies with block-wise channel mapping (i.e., Group Convolution) by varying group number $G$ and expansion ratio of hidden dimension $M$. For comparison, we use our AFBO with only CM by excluding the SM modules, and conduct experiments on ImageNet-1K dataset by employing DeiT-T and DeiT-S (Touvron et al., 2021a) as the basic backbones. As shown in Table 13, block-wise channel mapping is clearly inferior to our AFBO with only CM for similar model sizes. While increasing the hidden dimension can result in further performance improvements (along with an increase in parameters), block-wise channel mapping (Group Convolution) still lags behind our AFBO with only CM by about 1%~2%. These results clearly demonstrate that interaction between channel groups is crucial for effective channel mapping, and our AFBO module can efficiently facilitate information interaction among different groups.

Table 11: Comparison of AFBO with GLU by using expansion ratio of hidden dimension $M = 4$ on IN-1K and IN-R.

| Backbone | Method | Parameter | FLOPs | IN-1K Acc. | IN-R Acc. | Inference |
|---|---|---|---|---|---|---|
| Pool-S12 (Yu et al., 2022) | GLU ($M = 4$) | 16.6 M | 2.1 G | 78.5 % | 37.9% | 13.3 ms |
| | **AFBO (ours)** | **12.0 M** | **1.9 G** | **79.1 %** | **42.5%** | **11.2 ms** |
| Pool-S24 (Yu et al., 2022) | GLU ($M = 4$) | 30.9 M | 4.0 G | 80.5 % | 41.5 % | 19.7ms |
| | **AFBO (ours)** | **22.0 M** | **3.7 G** | **81.5 %** | **44.5%** | **16.4 ms** |
| Swin-T (Liu et al., 2021) | GLU ($M = 4$) | 36.9 M | 4.6 G | 81.3 % | 44.0 % | 19.8 ms |
| | **AFBO (ours)** | **27.5 M** | **4.4 G** | **82.1 %** | **45.8 %** | **15.5 ms** |

Table 12: Comparison of different backbones on GLUE benchmarks.

| Backbone | CoLA | RTE | MNLI | QNLI |
|---|---|---|---|---|
| GPT | 54.3 | 63.2 | 82.1 | 86.4 |
| GPT+AFBO | 56.8 | 65.3 | 82.7 | 86.9 |
| BERT-base | 54.8 | 67.2 | 83.5 | 90.1 |
| BERT+AFBO | 57.0 | 68.3 | 84.8 | 90.5 |
| BERT-large | 60.6 | 73.7 | 85.9 | 91.8 |
| BERT-large+AFBO | 61.2 | 74.8 | 86.5 | 92.2 |

Table 13: Comparisons of AFBO with block-wise channel mapping using DeiT-T and DeiT-S on IN-1K.

| Backbone | Method | $G$ | $M$ | Params. | Acc. |
|---|---|---|---|---|---|
| DeiT-T | Baseline | – | 4 | 5.7 M | 72.2% |
| | Group Convolution-A | 2 | 4 | 5.7 M | 70.8% |
| | Group Convolution-B | 4 | 4 | 5.0 M | 69.8% |
| | Group Convolution-C | 4 | 6 | 6.1 M | 71.7% |
| | **AFBO with only CM** | **–** | **4** | **5.7 M** | **73.7%** |
| DeiT-S | Baseline | – | 4 | 22.0 M | 79.8% |
| | Group Convolution-A | 2 | 4 | 22.0 M | 79.7% |
| | Group Convolution-C | 4 | 6 | 23.7 M | 80.0% |
| | **AFBO with only CM** | **–** | **4** | **21.7 M** | **80.8%** |

Table 14: Comparison of our AFBO and FULL channel mapping on IN-1K.

| Backbone | Method | Params (M) | Flops (G) | Latency (ms) | Accuracy (%) |
|---|---|---|---|---|---|
| DeiT-T | FULL | 7.7 | 1.6 | 13.8 | 74.7 |
| DeiT-T | AFBO | 6.0 | 1.3 | 13.2 | 74.6 |
| Swin-T | FULL | 36.9 | 4.6 | 19.8 | 81.3 |
| Swin-T | AFBO | 27.5 | 4.4 | 15.5 | 82.1 |

## A.5   COMPARISON WITH FULL CHANNEL MAPPING

We implement Eq. 4 as the FULL channel mapping method, and conduct additional experiments with the backbones of DeiT-T (Touvron et al., 2021a) and Swin-T (Liu et al., 2021) on IN-1K dataset. The results are listed in Table 14. For DeiT-T, the FULL implementation achieves slightly higher accuracy (74.7 % vs. 74.6 %) over our AFBO method at much higher cost of parameters (7.7 M vs. 6.0 M) and FLOPs (1.6 G vs. 1.3 G), and model latency (13.8 ms vs. 13.2 ms). In the case of Swin-T, our AFBO outperforms the FULL implementation by 0.8% in terms of accuracy while having much fewer parameters (27.5 M vs. 36.9 M) and model latency. Therefore, FULL implementation is limited to bring further accuracy improvement, but introduces much more computational cost.

## A.6   COMPARISON OF VIT WITH CHEBYSHEV KOLMOGOROV–ARNOLD NETWORKS (KAN)

In this subsection, we compare with ViT modified by KAN (Liu et al., 2025), whose implementation and results are publicly available at the website[1]. As shown in Table 15, our AFBO demonstrates superior performance across different DeiT (Touvron et al., 2021a) variants. Particularly, our AFBO is superior to KAN by 1.5%, 1.4%, and 1.0% on IN-1K with backbones of DeiT-T, DeiT-S, and DeiT-B, respectively. In terms of model robustness on OOD, AFBO also demonstrates superior performance on IN-R across different DeiT variants. Specifically, AFBO outperforms KAN by 4.8%, 3.6%, and 2.8% on IN-R with the backbones of DeiT-T, DeiT-S, and DeiT-B, respectively. These results verify the effectiveness of AFBO.

---

[1] https://github.com/snoop2head/KAN-ViT/blob/main/README.md

Table 15: Comparison of KAN and AFBO with DeiT backbones on IN-1K and IN-R.

| Backbone | Method | Accuracy on IN-1K | Accuracy on IN-R |
|----------|--------|-------------------|------------------|
| DeiT-T | KAN | 73.1% | 33.7% |
| DeiT-T | AFBO | 74.6% (**1.5 %**) | 38.5% (**4.8 %**) |
| DeiT-S | KAN | 79.7% | 41.7% |
| DeiT-S | AFBO | 81.1% (**1.4 %**) | 45.3% (**3.6 %**) |
| DeiT-B | KAN | 81.5% | 44.6% |
| DeiT-B | AFBO | 82.5% (**1.0 %**) | 47.4% (**2.8 %**) |

Table 16: Comparison of AFBO and the original FFN in terms of performance and CPU inference speed, where EdgeViT-xxs is used as backbone.

| Method | Params (M) | FLOPs (G) | CPU (ms) | IN-1K Acc. (%) |
|--------|-----------|-----------|----------|----------------|
| EdgeViT-xxs | 4.1 | 1.3 | 42.2 | 74.4 |
| EdgeViT-xxs + AFBO | 4.2 | 1.3 | 47.1 | 76.5 |

## A.7 COMPARISON OF FOR MOBILE MODEL

In this subsection, we conduct experiments by using EdgeViT (Chen et al., 2022) (i.e., a well-known lightweight model designed for mobile devices) and comparing it on an Intel(R) Xeon(R) Gold 5218 CPU @ 2.30GHz. As shown in Table 16, AFBO has comparable parameters and FLOPs while bringing 4.9 ms (about 10%) extra model latency. Particularly, our AFBO achieves a substantial accuracy improvement of 2.1%.

## A.8 GENERALIZATION ON NLP TASKS

We conduct experiments by applying AFBO to NLP tasks, where our AFBO is adopted to three widely used NLP models, including GPT (Radford, 2018), BERT-base (Devlin et al., 2019), and BERT-large (Devlin et al., 2019). All models are evaluated on four GLUE (Wang et al., 2019) benchmark tasks, i.e., CoLA, RTE, MNLI, and QNLI. As shown in Table 12, our AFBO brings consistent improvement over three original models across all tasks. Particularly, AFBO enhances the performance of GPT, BERT-base, and BERT-large with notable gains in CoLA and MNLI tasks. The results above clearly suggest that AFBO can be generalized well to the NLP domain.

Table 17: Comparisons with different models on INAT2019.

| Method | Parameter | Top-1 Acc. |
|--------|-----------|-----------|
| Swin-T (Liu et al., 2021) | 28.0 M | 71.0% |
| **+AFBO** | **27.5 M** | **72.7%** |
| Pyramid-VisionLLaMA-S (Chu et al., 2024) | 22.0 M | 68.4% |
| **+AFBO** | **22.1 M** | **69.2%** |
| Pyramid-VisionLLaMA-L (Chu et al., 2024) | 99.0 M | 74.5% |
| **+AFBO** | **99.5 M** | **75.1%** |

## A.9 CLASSIFICATION ON INATURALIST 2019

To further demonstrate the generalization ability of our AFBO, we transfer the pre-trained models to a long-tailed large-species classification task by using iNat2019 (Horn et al., 2018). Specifically, we fine-tune the pre-trained backbones of Swin Transformer (Liu et al., 2021) and VisionLLaMA (Chu et al., 2024) with our AFBO on iNat2019. As listed in Table 17, our AFBO consistently enhances the performance of these models. Specifically, AFBO improves the performance of Swin-T by 1.7% while reducing model size. Besides, AFBO achieves performance gains of 0.8% and 0.6% over the recently proposed Pyramid-VisionLLaMA-S and Pyramid-VisionLLaMA-L, respectively. These results clearly demonstrate the generalization ability of our AFBO on the challenging long-tailed large-species classification task.

### A.10 QUALITATIVE ANALYSIS OF AFBO

For qualitative analysis of our AFBO, we visualize the attention maps in the 12th block generated by the original Swin-T model and Swin-T with AFBO by using Grad-CAM (Selvaraju et al., 2017). The visualization results are shown in Fig 3 and Fig 4. From them we can see that our AFBO focuses on more discriminative and informative regions, and thus realizes high-accuracy classification on IN-1K. Furthermore, we compute the entropy for the features in the 12th block generated by the original Swin-T model and Swin-T+AFBO. As shown in Table 18, the feature entropy of Swin-T with AFBO is larger than the original Swin-T, which indicates that the features of Swin-T+AFBO are denser than ones of Swin-T, which is consistent with the visualization results. Therefore, features of Swin-T with AFBO involve more redundant information, which has proven very important for model robustness (Lin et al., 2024). It accounts for the clear improvement brought by our AFBO on OOD benchmarks.

Table 18: Comparison of AFBO and the original FFN in terms of performance and feature entropy, where Swin-T is used as backbone.

| Model | Params. (M) | Flops (G) | IN-1K (%) | Feature Entropy | IN-A (↑%) | IN-R (↑%) | IN-SK (↑%) | IN-C (↓%) |
|---|---|---|---|---|---|---|---|---|
| Swin-T | 28.0 | 4.6 | 81.2 | 2.7 | 21.7 | 41.3 | 29.0 | 62.0 |
| Swin-T+AFBO | 27.5 | 4.4 | 82.1 | 4.7 | 26.0 | 45.8 | 31.7 | 56.4 |

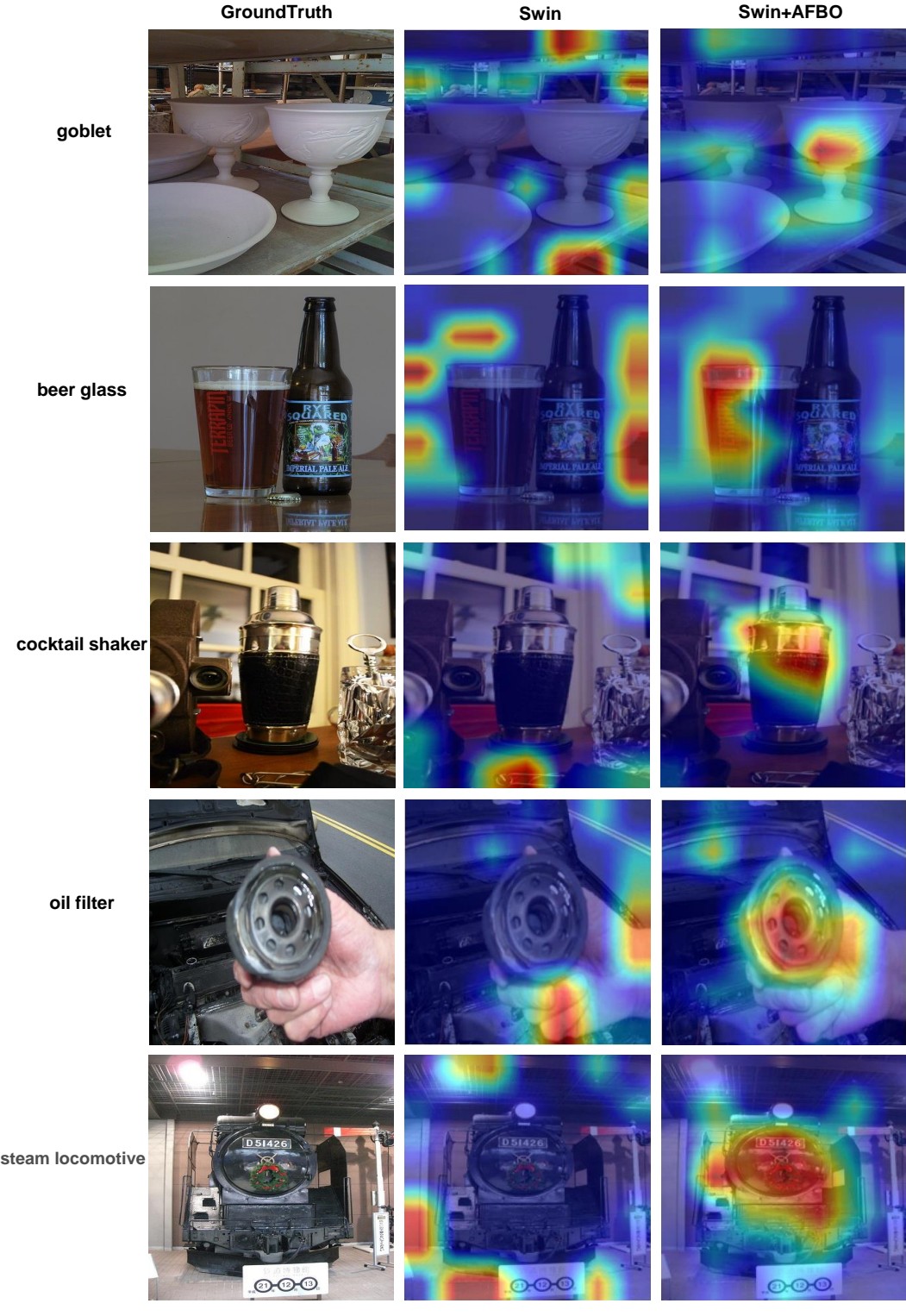

Figure 3: Visualization of attention maps in the 12th block generated by the original Swin-T model and Swin-T with AFBO by using Grad-CAM (Selvaraju et al., 2017).

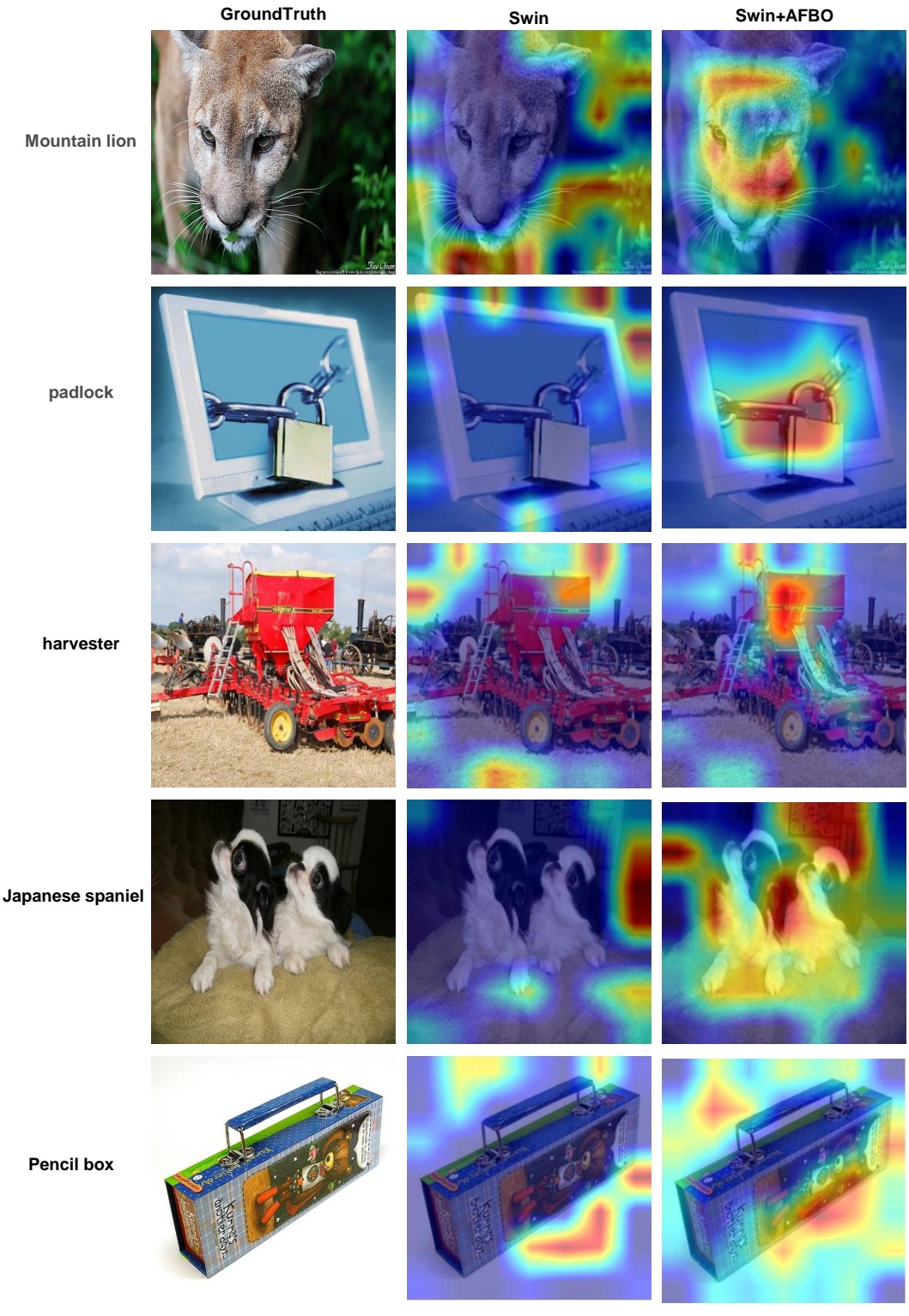

Figure 4: Visualization of attention maps in the 12th block generated by the original Swin-T model and Swin-T with AFBO by using Grad-CAM (Selvaraju et al., 2017).

