# OpenReview forum: "Asymmetric Factorized Bilinear Operation for Vision Transformer"
_ICLR.cc/2025/Conference — ICLR 2025 Poster_

### Official Review · Reviewer_QMDZ · 2024-10-30

**Soundness:** 2
**Presentation:** 2
**Contribution:** 2
**Rating:** 5
**Confidence:** 4

**Summary:**

This paper introduces an Asymmetric Factorized Bilinear Operation (AFBO) as a replacement for the  traditional channel-mixer (FFN/MLP) in ViT architectures, with the goal of improving accuracy-complexity tradeoff. The AFBO first employs a spatial-channel factorized bilinear operation to capture second-order statistics of the input features. To manage computational load, the authors also develop two structured-sparsity channel mapping strategies (OCCM and GCCM), which preprocess features prior to the factorized  bilinear operation in AFBO. The experimental results show that AFBO has better performance than traditional FFN/MLP module in various ViT and CNN-ViT hybrid architectures,  and is competitive to its counterparts.

**Strengths:**

1. The proposed AFBO utilizes a factorized bilinear approach [1] to enhance FFN performance, presenting a relatively new perspective for channel-mixing mechanisms.

2. The motivation of AFBO are clearly conveyed, with the Factorized Bilinear Operation thoroughly explained. Additionally, the authors provide visualizations of their channel mapping strategies.

3. The comparative experiments are well-structured and extensive. The authors conduct replace-and-play experiments across 20 ViT-like architectures of varying scales, demonstrating the utility and impact of the proposed AFBO.

[1] Li Y, Wang N, Liu J, et al. Factorized bilinear models for image recognition[C]//Proceedings of the IEEE international conference on computer vision. 2017: 2079-2087.

**Weaknesses:**

1. While the authors provide theoretical complexity metrics (e.g., FLOPs and parameter counts), the on-device speed of the proposed components is not well illustrated. Metrics such as latency and throughput, which are widely adopted for evaluating actual operating efficiency, are not fully reported. In Table 1, the authors include limited latency results, showing that the proposed method has slower runtime compared to the original FFN and GLU. However, the specific configurations for these latency comparisons are unclear, and on-device speed metrics are omitted for the all remaining experiments. These makes it difficult to evaluate the efficiency contribution of the proposed method.

2. The AFBO framework introduces two complex channel mapping schemes to reduce the effective channel width during mixing, leveraging feature redundancy. However, instead of these complex mappings, why not explore a simpler partial-channel projection approach, as commonly used for complexity reduction [2,3]? The paper lacks experiments demonstrating that the proposed channel mapping strategies outperform this simpler method.

3. Overall, the AFBO design is intricate, which may hinder fast inference. The AFBO framework employs two complex channel mapping operations with multiple linear layers, along with split-concatenate operations for feature preprocessing, followed by two depth-wise convolutions for spatial modeling and a SiLU activation function. Both depth-wise convolution and SiLU activation are known to have good theoretical efficiency but may perform slowly in practice. In contrast, the original FFN employs only two linear layers and a single activation function, highlighting a potential disadvantage in AFBO's real running speed.

4. The authors claim that the Factorized Bilinear Operation better captures second-order statistics than traditional FFN. However, the paper lacks intuitive visualizations that could illustrate the improved feature modeling achieved by AFBO over FFN, as well as theoretical/statistical comparisons of features learned by both methods. This weakens the claimed contributions of the Factorized Bilinear Operation.

5. Compared to the main experiments, the ablation studies lack clarity. For example, in Table 5, the paper presents several AFBO variants, but the structural details of each variant are unclear. In the AFBO-S configuration, which uses only spatial modeling, it is unclear whether channels are split into two parts for processing by two depth-wise convolutions or treated as a single unit. Similarly, in AFBO-C, which employs only channel modeling, it is unclear if different channel mappings involve activation and multiplication or if features pass directly to the final linear projection. Furthermore, there are no experiments directly validating the effectiveness of the Factorized Bilinear Operation, raising questions about whether the accuracy gains stem from the added depth-wise convolutions for spatial encoding rather than the bilinear operation itself.

6. The detailed configurations of the proposed AFBO are not presented clearly. Readers may need to re-read sections multiple times to fully grasp the architecture. For instance, in Figure 1, feature dimensions are not annotated, making it challenging to understand each operation's role. In part (b) of Figure 1, after the left spatial modeling (SM) module, the use of product and positional embedding-like markers is unclear, with no definition provided in the figure or caption. It appears that positional embedding is applied in AFBO, which seems incongruous with the methodological statements.

7. This paper defines the proposed methods and variants using numerous long and similar abbreviations (e.g., AFBO, SCFBO, OCCM, GCCM, SCFBO-GC, SCFBO-OC, AFBO-C, AFBO-S, AFBO-CS, AFBO-SC), which are difficult to remember and require repeated reading for clarity.

In conclusion, although this paper demonstrates FFN improvements in its comparative experiments, there is insufficient explanation for the mechanism behind these improvements, and some figures and tables lack clarity. The experiments do not offer direct evidence that the factorized bilinear operation and channel mapping schemes are truly effective compared to simpler alternatives. Moreover, the omission of on-device speed comparisons, with only theoretical complexity provided, raises questions about the method's actual efficiency improvements in practical settings. It is recommended that the authors address these concerns to strengthen the paper.

[2] Chen J, Kao S, He H, et al. Run, don't walk: chasing higher FLOPS for faster neural networks[C]//Proceedings of the IEEE/CVF conference on computer vision and pattern recognition. 2023: 12021-12031.

[3] Han K, Wang Y, Tian Q, et al. Ghostnet: More features from cheap operations[C]//Proceedings of the IEEE/CVF conference on computer vision and pattern recognition. 2020: 1580-1589.

**Questions:**

For primary concerns, please refer to the weaknesses noted above. Below are several additional minor questions:

1. Given that ReLU activation is considerably faster than SiLU/GeLU and widely used, why did the authors not consider ReLU activation? Also, providing results with ReLU in Table 6 would offer a clearer comparison.

2. What are the specific configurations for feature-level transformations during the proposed channel mapping operations? Are these operations implemented through split-concat operations, in-feature transformations, or masking? Variations in these implementations could lead to significant differences in practical speed and CUDA compatibility.

3. Since the authors emphasize optimizing the accuracy-efficiency trade-off as a primary contribution, is the proposed method operable on mobile devices? Given the reliance on self-defined feature-level transformations, can these operations be effectively exported to ONNX formats?

---

### Official Review · Reviewer_HXbZ · 2024-11-01

**Soundness:** 3
**Presentation:** 3
**Contribution:** 2
**Rating:** 6
**Confidence:** 4

**Summary:**

The paper introduces a novel operation, AFBO, designed to replace the feed-forward network (FFN) in vision transformers (ViTs). The core contribution is the proposal of a spatial-channel factorized bilinear operation that models second-order statistics of token features, aiming to enhance feature learning while reducing computational complexity. The paper also presents two structured-sparsity channel mapping strategies, GCCM and OCCM, to further decrease model complexity. Experiments conducted across various tasks and models demonstrate the effectiveness of AFBO in improving performance and complexity trade-offs.

**Strengths:**

The paper introduces a novel AFBO operation that serves as a more useful component for ViT visual tasks.
The experiments are thorough, containing most visual backbones and visual tasks.
The paper is well-written with rather clear explaination.

**Weaknesses:**

The novelty of this work is rather limited as only FFN’s first layer is conducted change with spatial and channel attention which are normal methods for feature transformation.
While the paper demonstrates the effectiveness of AFBO, there could be more discussion on the theoretical underpinnings of why the strategies of AFBO work well.
The performance improvement of the proposed method is relatively insufficient, and it also brings additional computational costs.

**Questions:**

1.	Although the paper focuses on vision tasks, a brief discussion on how AFBO might generalize to other domains, such as natural language processing and multimodal representation learning, would be beneficial. It would be better to provide experiments on popular multimodal models to prove its effectiveness.
2.	I have a few more questions on the reasons behind the design of channel attention. Why using OCCM on the on side with activation and GCCM on the other side? How will the number of Group G1 and G2 affect the results for OCCM and GCCM and how do you choose these hyperparameters as ablation study and discussion is missing.
3.	As AFBO aims to introduce the spatial correlation of CNN into ViT, should there be more significant performance gain in pure ViT(such as DeiT) than transformers considering similar correlations(such as Swin-T), and furthermore, is this structure powerful enough to introduce 2-d correlations so that there is no need for improvement in attention module before.
4.	 Authors are recommended to discuss and compare other FFN modification methods, like KAN.

---

### Official Review · Reviewer_vUSe · 2024-11-02

**Soundness:** 4
**Presentation:** 4
**Contribution:** 3
**Rating:** 8
**Confidence:** 4

**Summary:**

The paper introduces the Asymmetric Factorized Bilinear Operation (AFBO), an approach to replace the feed-forward network (FFN) in Vision Transformers (ViTs). AFBO aims to improve the feature learning ability of ViTs by efficiently exploring rich statistics of token features, leading to better performance and complexity trade-offs. The paper proposed SCFBO,  a method to compute second-order statistics via a factorized bilinear operation, replacing the simple linear projection in FFN. Two strategies, Grouped Cross Channel Mapping (GCCM) and Overlapped Cycle Channel Mapping (OCCM), are introduced to reduce computational complexity while maintaining model performance. Experiments conducted on various vision tasks using different ViT models demonstrate AFBO's superiority in terms of generalization and robustness.

**Strengths:**

1. AFBO improves ViTs by modeling second-order correlations among token features, leading to better feature representations.
2. The structured-sparsity channel mapping strategies reduce computational complexity, making AFBO more efficient.
3. AFBO achieves comparable or better performance than existing FFN variants on various vision tasks.

**Weaknesses:**

I would recommend this paper to be accepted. I only have some questions:
1. Could the authors provide some visualization of feature interactions to demonstrate the effectiveness of the proposed method.
2. Why use the DW conv to replace the original conv for spatial modeling, could the authors discuss this more specifically?

**Questions:**

1. I would suggest the authors to simplify the second paragraph in Introduction, highlighting the motivation more directly.
2. A minor in Figure 1, Line 074, the period.

---

### Official Review · Reviewer_QBni · 2024-11-03

**Soundness:** 2
**Presentation:** 2
**Contribution:** 2
**Rating:** 6
**Confidence:** 5

**Summary:**

The paper proposes an Asymmetric Factorized Bilinear Operation block as a drop-in replacement for the FFN in Vision Transformers. It consists of a Grouped Cross Channel Mapping and Overlapped Cycle Channel Mapping to perform cross-channel interactions between channel groups to improve the performance of ViTs. Experiments are conducted on ImageNet variants, Object Detection and Instance Segmentation tasks across different architectures.

**Strengths:**

The proposed AFBO block introduces two new modules, GCCM and OCCM, to enhance ViT models performance. It is compatible with various ViT models, and extensive experiments across different ViT architectures and tasks demonstrate the method's effectiveness.

**Weaknesses:**

Although the proposed method is inspired by bilinear models, the final implementation seems engineered to balance accuracy and efficiency. Particularly, L237-244 mentions that SiLU, GELU activation functions cause a performance drop and use this to motivate the design of the proposed AFBO module. Did the authors experiment with other activation functions like ReLU?

Missing inference speed analysis and comparison with FFN and AFBO. Table 1 only shows the latency of AFBO block but not the inference speed of the full model. Group convolutions are known to be slower. Table 1 already shows that AFBO is ~2ms slower than the standard FFN and so the overall speed might be significantly slower than the baseline model. Can the authors provide the full inference speed analysis?

The GCCM and OCCM modules contain group numbers G1 and G2 that are fixed. The authors do not discuss how to determine the optimal number of groups. Different tasks or datasets may have different optimal group numbers.

Missing FLOPs for Table 4.

Although the paper shows experiments with multiple ViTs, it is missing the important baseline of the standard ViT backbone. It is also missing the ablation experiment of implementing equation (4) in a standard way. Although it is expensive, this experiment will show the performance difference when using the proposed GCCM and OCCM modules instead of the naïve implementation.

Missing qualitative analysis and comparison when using the proposed AFBO module.

**Questions:**

Table 2 shows that the proposed method is not very effective when scaling up the model as the performance improvements diminish.
If the hyperparameters for AFBO are fixed, why do the parameters/FLOPs increase for some and decrease for other models in Table 4?

Although Table 4 shows that the proposed method works for object detection and segmentation, it also increases the parameters and FLOPs. It makes it hard to compare. Can increasing the parameters and FLOPs of the baseline model also achieve similar improvement?

---

### Meta-Review · Area_Chair_QN7b · 2024-12-23

**Metareview:**

The paper proposes an Asymmetric Factorized Bilinear Operation block as a drop-in replacement for the FFN in Vision Transformers. AFBO aims to improve the feature learning ability of ViTs by efficiently exploring rich statistics of token features, leading to better performance and complexity trade-offs.  Experiments conducted on various vision tasks using different ViT models demonstrate AFBO's superiority in terms of generalization and robustness. All reviewers appreciated the effect of the new channel-mixing mechanism and its consistent performance over different ViT architectures and tasks. The main concerns were unconvincing technical novelty, unclear exposition, and missing analyses/comparisons. The authors provided detailed point-to-point rebuttals with more experiments, addressing most concerns. AC  thus recommends acceptance.

**Additional Comments On Reviewer Discussion:**

The main concerns were unconvincing technical novelty, unclear exposition, and missing analyses/comparisons. After the rebuttal, two reviewers (vUSe and QBni) actively engaged in the discussion and confirmed their positive scores. Reviewer HXbZ, who raised the concern about the novelty, also increased his/her score to borderline accept, acknowledging the authors' rebuttal.

---

### Decision · Program_Chairs · 2025-01-22

Accept (Poster)